# Protein Phosphatase 2A as a Therapeutic Target in Pulmonary Diseases

**DOI:** 10.3390/medicina59091552

**Published:** 2023-08-26

**Authors:** Howard Yu, Sahil Zaveri, Zeeshan Sattar, Michael Schaible, Brais Perez Gandara, Anwar Uddin, Lucas R. McGarvey, Michael Ohlmeyer, Patrick Geraghty

**Affiliations:** 1Department of Medicine, State University of New York Downstate Health Sciences University, 450 Clarkson Avenue, Brooklyn, NY 11203, USA; howard.yu@downstate.edu (H.Y.); sahil.zaveri@downstate.edu (S.Z.); zsattar@kumc.edu (Z.S.); michael.schaible@downstate.edu (M.S.); brais.perezgandara@downstate.edu (B.P.G.); anwar.uddin@downstate.edu (A.U.); panda.ant635@gmail.com (L.R.M.); 2Atux Iskay LLC, Plainsboro, NJ 08536, USA; michael.ohlmeyer@gmail.com

**Keywords:** PP2A, lung diseases, inflammation, future therapy

## Abstract

New disease targets and medicinal chemistry approaches are urgently needed to develop novel therapeutic strategies for treating pulmonary diseases. Emerging evidence suggests that reduced activity of protein phosphatase 2A (PP2A), a complex heterotrimeric enzyme that regulates dephosphorylation of serine and threonine residues from many proteins, is observed in multiple pulmonary diseases, including lung cancer, smoke-induced chronic obstructive pulmonary disease, alpha-1 antitrypsin deficiency, asthma, and idiopathic pulmonary fibrosis. Loss of PP2A responses is linked to many mechanisms associated with disease progressions, such as senescence, proliferation, inflammation, corticosteroid resistance, enhanced protease responses, and mRNA stability. Therefore, chemical restoration of PP2A may represent a novel treatment for these diseases. This review outlines the potential impact of reduced PP2A activity in pulmonary diseases, endogenous and exogenous inhibitors of PP2A, details the possible PP2A-dependent mechanisms observed in these conditions, and outlines potential therapeutic strategies for treatment. Substantial medicinal chemistry efforts are underway to develop therapeutics targeting PP2A activity. The development of specific activators of PP2A that selectively target PP2A holoenzymes could improve our understanding of the function of PP2A in pulmonary diseases. This may lead to the development of therapeutics for restoring normal PP2A responses within the lung.

## 1. Introduction

Chronic respiratory disorders affect the lungs, airways, and other structural components, leading to widespread illness and mortality. Lung cancer, asthma, chronic obstructive pulmonary disease (COPD), and idiopathic pulmonary fibrosis (IPF) are some of these respiratory conditions, and are among the top causes of morbidity and mortality worldwide. From 1990 to 2017, there was an increase in the prevalence of chronic lung disease worldwide. In 2017, 544.9 million individuals worldwide were impacted by chronic respiratory diseases, which is a rise of 39.8% from the 1990 data [1]. This equated to a global prevalence of 7.1% [1].

The most common lung cancer types, non-small cell lung cancer (NSCLC) and small cell lung cancer (SCLC), account for approximately 80–85% and 10–15% of all instances of lung cancer, respectively [2,3]. The 5-year survival rate for patients with lung cancer is presently only 18.6%, which is significantly lower compared to other common forms of cancer. Lung cancer is among the primary causes of death across the globe, resulting in over 1.3 million fatalities each year [4]. Recent data indicate that the number of people dying from lung cancer worldwide is expected to rise significantly by 2035, reaching 3 million. This increase will affect both men and women, with deaths in men projected to double from 1.1 million in 2012 to 2.1 million in 2035 and deaths in women projected to double from 0.5 million in 2012 to 0.9 million in 2035 [5]. Moreover, lung cancer is more prevalent in individuals with COPD than in those without it [4], and smoking is the most common connection between COPD and lung cancer [6]. The number of people worldwide with COPD is predicted to be 328 million [1,5,6]. COPD causes persistent shortness of breath and limited lung airflow; COPD is degenerative and tends to worsen over time. While medication can relieve symptoms, it cannot stop or reverse the disease’s progression. COPD can be classified into two primary conditions: bronchitis and emphysema. Both of these eventually lead to chronic dyspnea. Although smoking causes both COPD and lung cancer, their relationship is not solely due to smoking, as COPD is an independent risk factor for lung cancer [7]. Several overlapping molecular pathways are observed in COPD and lung cancer but also in other pulmonary diseases, such as asthma and IPF. The incidence and mortality rate of asthma has decreased over the past years [8]. While asthma prevalence and mortality rates have decreased, managing varying levels of asthma severity is challenging due to corticosteroid sensitivity [9]. The frequency of IPF is increasing [10], with the estimated occurrence rates of IPF increasing from approximately 3 to 9 cases per 100,000 individuals annually in Europe and North America between 1998 and 2012 [11].

One of the common molecular players observed to be altered in all of these pulmonary diseases is the serine/threonine (Ser/Thr) phosphatase, protein phosphatase (PP) 2A (PP2A). This review will outline the activation status of PP2A in pulmonary diseases and the subsequent impact on multiple pathways, discuss the potential means of targeting PP2A responses, and possible outcomes of targeting PP2A activity systemically.

## 2. Protein Phosphatase 2A

Reversible phosphorylation of signal mediators is a cornerstone of intracellular signaling and is primarily regulated by a delicate balance of intracellular and extracellular kinases and phosphatases [12]. Phosphoprotein phosphatase is the largest Ser/Thr phosphatase family comprising several members, including PP1, PP2A, PP2B, and PP4. PP2A constitutes about 1% of the total cellular protein content and, along with PP1, is responsible for more than 90% of all Ser/Thr phosphatase activity within the cell [13]. PP2A holoenzyme is composed of three discrete subunits. These are the scaffolding (A), regulatory (B), and catalytic (C) subunits. The A and C subunits each have two possible variations, α and β [14,15,16]. The B subunit exists as four different classes, each with 2–5 isoforms. This leads to numerous combinations of the PP2A holoenzyme, thus imparting specificity for different targets (see Figure 1). The main pathways influenced by PP2A are the PI3K (Akt), mTOR (p70S6K), and MAPK (MEK/ERK) pathways, but PP2A also targets pathways such as cMyc, Wnt signaling (GSK-3β and β-catenin), apoptosis (Bcl2, Bad, and FOXO proteins), tau signaling, cell cycle regulation (cdc25, WEE1, and pRb), and DNA damage responses (p53, ATM, and Chk) [17,18,19,20,21,22]. The canonical B-subunit picture is incomplete with four recent papers from independent research groups adding to it: Huang et al., identified that PP2A AC binds to the Integrator-RNAPII complex [23], cryo-EM structures of the PP2A AC-Integrator complex were recently determined that provide a structural basis for understanding the PP2A AC-Integrator transcriptional regulation [24,25], and binding of the PP2A AC to the integrator complex subunits INTS6 and INTS8 subunit to regulate transcription [26]. The effect of PP2A in this context is to enhance promoter proximal stalling of RNAPolII by dephosphorylation of its C Terminal Domain (CTD), and restrain gene transcription. Due to the pleiotropic nature of PP2A actions, it regulates many biological processes such as cell proliferation, cell survival, gene transcription, protein synthesis, cell migration, and invasion [27].

### PP2A Subunits

The A subunit exists in two isoforms, PR65/Aα and PR65/Aβ, encoded by the *PPP2R1A* and *PPP2R1B* genes, respectively. This particular subunit serves as a backbone for holoenzyme recombination [27]. These isoforms are characterized by repeats of amino acids organized into two anti-parallel α-helices referred to as HEAT repeat domain. HEAT repeats are named after the first four proteins found to contain them: Huntington, EF3, PP2A A subunit, and TOR1. These repeats are arranged in a crescent-like configuration in the PP2A A subunit [28].

Similar to the A subunit, the C subunit has two isoforms that serve as PP2A holoenzyme’s catalytic domain. The *PPP2CA* and *PPP2CB* genes encode these isoforms, each forming a globular structure folding on itself in an α/β arrangement. *PPP2CB* has weaker promoter activity, resulting in its expression being ten-fold lower than *PPP2CA* [27]. It is transcribed in an inactive form and is post-translationally activated after the assemblage of the entire PP2A holoenzyme [29,30]. The catalytic C subunit binds to the HEAT repeats of the A subunit at the 11–15 positions [31]. The catalytic function of this subunit is brought about by binding two manganese atoms to phosphate, thus facilitating the hydrolysis of serine/threonine phosphate esters. The resulting AC dimer is also known as the core enzyme when attached to the A subunit [32].

B subunit isoforms bind in a mutually exclusive manner, providing subcellular localization, substrate specificity, and physiologic function of the final PP2A product [33,34]. As mentioned above, the B subunit exists in four different classes. These are: B (B55/PR55), B’ (B56/PR61), B’’ (PR48/PR72/PR130), and B’’’ (PR93/PR110)/striatin [35]. The B (B55/PR55) class has five different isoforms (α, β, β2, γ, and δ), which are encoded by four genes (*PPP2R2A*, *PPP2R2B*, *PPP2R2C*, and *PPP2R2D*) [36,37,38]. The B’ (B56/PR61) class is the largest subtype family containing ten isoforms (α, β, γ, δ, δ, ε, ε-s) and are encoded by five genes *PPP2R5* (*A*-*E*) [12,13,14,39,40]. The B’’ (PR48/PR72/PR130) class is encoded by three genes (*PPP2R3A*, *PPP2R3B*, and *PPP2R3C*) comprising of six isoforms [41]. B‴ (PR93/PR110)/striatin subunits are encoded by three genes (*STRN*, *STRN3*, and *STRN4*), each producing one isoform. The combination of all the different subunits, their subclasses, and isoforms leads to more than 80 different PP2A complexes, each with unique physiological functions. The B subunit attaches to repeats 1–10, while the C subunit occupies repeats 11–15 [42].

## 3. The Activation Status of PP2A in Pulmonary Diseases

Impairment of PP2A function is commonly observed in many cancers, but it also occurs in other diseases and, prominently, within lung diseases. Multiple mechanisms, including genetic and post-translational modifications, are reported to impact PP2A function and activity in pulmonary diseases. New research has shown that inhibiting PP2A can contribute to the pathology of COPD, alpha-1 antitrypsin (AAT) deficiency, lung cancer, asthma, and IPF [43,44,45,46,47,48,49,50,51]. This section will examine the possibility that reduced PP2A activity could be a unifying factor in several pulmonary diseases (see Table 1). Exploring the molecular mechanisms responsible for alterations of PP2A activation could be an essential step in pulmonary disease pathology.

### 3.1. PP2A in Cigarette Smoke-Induced COPD

PP2A plays a major regulatory role in initiating and maintaining inflammatory disease seen in COPD [72]. Modulation of PP2A activity downregulates cytokine expression and prevents the induction of proteases following smoke exposure [43,44,54]. Conversely, the presence of a PP2A inhibitor, okadaic acid, or direct inhibition of PP2A expression enhances smoke-induced inflammation in mice [43,54]. Mechanistically, PP2A can mediate the dephosphorylation of IκB, which results in the stabilization of IκB and the subsequent inhibition of NFκB, a significant mediator of both acute and chronic inflammatory processes [73]. Furthermore, PP2A negatively regulates the transcriptionally active NFκB by dephosphorylating its RelA (p65) subunit [74]. Diminished PP2A activity is observed to occur in parallel with increased phosphorylation of MAPK, ERK, activation of NFκB, increased airspace enlargements, and loss of lung function in smoke-exposed mice [43,44,45,52,54].

Cigarette smoke enhances endogenous inhibitors to suppress PP2A and contributes to suppressing PP2A activators, such as protein tyrosine phosphatase 1B (PTP1B) [45,52]. PP2A becomes inactivated by phosphorylation of a tyrosine site at position 307 (Tyr307) in the catalytic subunit of the enzyme, which is offset by the activity of PTP1B [52]. We recently demonstrated enhanced lung damage in *Ptp1b* deficient mice when exposed to cigarette smoke [75]. Smoke inhalation also subdues PTP1B activity, and smoke-exposed *Ptp1b* deficient mice are more susceptible to COPD [52,75,76]. Smoke-induced TLR9 expression results in TLR9 directly binding and inactivating PTP1B [76].

A critical contributor to COPD disease pathogenesis is maladaptive airway remodeling. While the role of serine elastase and matrix metalloproteinases in the pathogenesis of COPD is rigorously described, there is emerging interest in the role of the cathepsin family of enzymes as a major initiator of pathologic airway remodeling [77]. Cathepsin S is a lysosomal peptidase and member of the C1 family of cysteine proteases, which contains cathepsin E, G, and K [43]. Cathepsin S is unique amongst others in the cathepsin family due to its ability to function in a pH neutral environment which can potentially contribute to early COPD disease development [78]. Our group demonstrated that PP2A activation negatively regulates cathepsin S expression at a transcriptional level for the first time [43]. Thus, this suppresses early deleterious airway changes in the presence of cigarette smoke.

PP2A also plays a crucial role in mitigating and resolving established inflammatory responses via regulating the mRNA-destabilizing protein tristetraprolin (TTP). Enhancing TTP activity reduces the severity of cigarette smoke-induced experimental COPD [55]. After smoke-induced inflammation, basal PP2A activity dephosphorylates TTP, activating it to destabilize the mRNA of proinflammatory cytokines and chemokines in inflammatory cells at the site of injury or infection and contributes to the resolution of the inflammatory response [79].

### 3.2. PP2A Responses in Alpha-1 Antitrypsin Deficiency

The development of COPD in AAT deficiency individuals is the best described genetic associated link to COPD development. By examining PP2A activity levels in neutrophils from healthy MM and AAT deficiency ZZ individuals, our group observed diminished PP2A in the neutrophils from AAT deficiency subjects. Additionally, AAT supplementation in AAT deficiency individuals increases PP2A activity in neutrophils, monocytes, small airway epithelial cells, and A549 cells. Mechanistically, AAT increased PP2A activity in a PTP1B-dependent manner in vitro and in vivo. AAT stimulation leads to PP2A dephosphorylation and activation at Tyr307 in the catalytic C subunit, a specific target of PTP1B phosphatase [45]. Furthermore, AAT can activate PP2A via protein kinase A (PKA) induction, which mediates the phosphorylation of the B subunit PPP2R5D to enhance PP2A activity [80]. As a result of augmented PP2A activation by AAT, BALF samples from AAT deficiency individuals undergoing supplementation AAT therapy had significantly reduced levels of MMP-1, MMP-9, IL-8, IL-1β, MCP-1, and TNF-α. Therefore, loss of PP2A responses in AAT deficiency alters the protease and inflammation profile in the lungs.

### 3.3. PP2A in Asthma

Yoshiki and colleagues observed that severe asthmatic patients who are treatment resistant to conventional steroid therapy had increased phosphorylation of JNK-1 and glucocorticoid receptor (GR) Ser226 with concomitant decreases in PP2A activity [66]. IFN-γ/LPS induce miR-9 expression reduces PP2A regulatory subunit B (B56) δ isoform, altering PP2A activity and inhibiting dexamethasone-induced GR nuclear translocation [81]. Eosinophil peroxidase enhances PP2A phosphorylation, leading to reduced PP2A protein expression and activity [67]. This inactivation can be reversed by formoterol or omalizumab, a monoclonal antibody that inhibits cell degranulation [67,68]. PP2A reduction by siRNA enhances CCL4, IL-13, and iNOS expression in eosinophils co-cultured with airway epithelial cells [68]. Similar to human data, PP2A activity is reduced in the ovalbumin mouse model of asthma [69]. Activating PP2A with FTY720 (Fingolimod/ Fingolimod Hydrochloride, a sphingosine-1-phosphate (S1P) antagonist) suppresses tissue inflammation and serum IgE levels in the ovalbumin-induced asthma mouse model, which is further enhanced when combined with proteasome inhibitor bortezomib, which suppresses CIP2A expression by an unknown mechanism [65]. Equally, the phosphodiesterase inhibitor, theophylline, enhanced PP2A responses to counter inflammation in A549 cells [82]. Theophylline is known to relax bronchial smooth muscle and is used in the treatment of asthma and COPD [83]. Salmeterol is also reported to induce PP2A activity in a house dust mite (HDM) and rhinovirus 1B infection mouse model [84]. PP2A activity is also known to play a role in mast cell degranulation targeting the TNF-related apoptosis-inducing ligand (TRAIL) with 2-amino-4-(4-heptyloyphenol)-2-methylbutan-1-ol (AALs) is reported to increase PP2A in allergic mice that subsequently reduced eosinophilia, TGF-β1, and peribronchial fibrosis [51,85].

Patients with severe eosinophilic chronic rhinosinusitis characterized by local resistance to corticosteroids had decreased PP2A mRNA expression in the nasal epithelium, which was improved by PP2A plasmid transduction, omalizumab, and long-acting beta agonist [67,68]. Native, non-carboxymethylated, PP2A trimeric enzyme is bounded by the B′δ regulatory B subunit, which maintains p38 and MK2 in the inactive, non-phosphorylated state, preventing degranulation. However, with increased antigen binding of the FcεRI receptors during an asthma exacerbation, PP2A is carboxymethylated, leading to the exchange of B′δ for the Bα subunit. This allows for the phosphorylation and activation of p38, MK2, and MAPK leading to degranulation and disease [86].

### 3.4. PP2A in Pulmonary Fibrosis

The deletion of PP2A is linked to the pathogenesis of IPF, as PP2A activation responses are blunted in fibroblasts isolated from IPF subjects [50]. Myeloid-specific deletion of *Ppp2r1a* (the gene responsible for subunit A) transgenic mouse models exposed to ambient particulate matter 2.5 (PM_2.5_) resulted in increased inflammation and pulmonary fibrosis in these animals [70]. PM_2.5_ exposure can induce oxidative damage, fibrosis, and collagen deposition [87]. Similarly, Sun et al. 2019 demonstrated that myeloid-specific deletion of PP2A in murine models increased lung fibrosis and matrix collagen deposition when challenged with bleomycin [71]. Direct β-catenin activation contributes to the induction of epithelial-mesenchymal transition (EMT), a critical process in tissue fibrosis and repair after injury [88,89,90,91,92,93,94,95]. α2β1 integrin is an extracellular receptor found on the surface membrane of pulmonary fibroblasts. α2β1 integrin receptor promotes dephosphorylation and activation of GSK-3β through PP2A [96]. In IPF, low α_2_β_1_ integrin receptor concentrations lead to decreased PP2A levels, increased phosphorylated GSK-3β, and increased β-catenin and fibroblast proliferation.

Tissue analysis of eight IPF patients’ lung biopsies revealed elevated MID1 and TRAIL expression and decreased PP2A activity [50]. To further elucidate the role of PP2A, TRAIL knockout mice administered bleomycin or animals treated with the PP2A activator FTY-720 reduced bleomycin-induced fibrosis, and reduced expression levels of profibrotic genes, reduced collagen deposition, and decreased apoptosis. Importantly, TRAIL knockout animals and FTY-720 treated mice were protected from bleomycin-induced fibrosis as observed by preserving normal levels of vital capacity and compliance as seen during pulmonary function testing [51]. Theophylline may have some potential in the treatment of IPF as it can inhibit TGFβ -mediated transition of pulmonary fibroblasts into myofibroblasts by regulaing the cAMP-PKA pathway and theophylline suppresses COL1 gene expression [97]. A recent study demonstrated that theophylline could prevent bleomycin induced pulmonary fibrosis in mice by inhibiting Th17 differentiation and TGFβ signaling [98].

### 3.5. PP2A in Lung Cancer

Loss of PP2A subunits via gene mutations or its inhibition by various endogenous inhibitors is associated with the development of various cancers [56]. Particularly, mutations in the PP2A-Aα subunit correlate with lung cancer (LC), occurring in 1.3% of LC tumors [99]. Furthermore, PP2A-Aβ mutation results in dysfunction of downstream GTPase RalA. PP2A dephosphorylation of RalA leads to tumor suppression and loss of RalA regulation, resulting in dysfunctional cell growth, migration, and apoptosis [57].

Mutations in genes encoding PP2A’s regulatory β-subunit are also implicated in lung adenocarcinoma cell proliferation. The PP2A-B56γ isoform is responsible for dephosphorylation and inactivation of ERK and thus aborting cancer cell growth and tissue invasion [58,59]. c-MYC transcription factor signaling significantly promotes NSCLC tumorigenesis and cancer cell metabolism [100]. When a B56α regulatory subunit binds the PP2A holoenzyme, it provides targeting specificity for c-MYC. PP2A can then directly inactivate c-MYC via dephosphorylation at serine 62, effectively neutralizing its pro-oncogenic activity [60]. Additionally, constitutive activation of Akt serine/threonine kinase is critical in activating pro-oncotic gene expression, promoting NSCLC cell survival despite chemotherapy and radiation [61,62,63]. PP2A activation, via endogenous inhibitor suppression, positively correlates with reduced phosphorylated Akt and subsequent increased cancer cell death [101,102]. The upregulation of the PP2A inhibitor CIP2A is associated with increased cancer cell proliferation and the upregulation of multiple downstream mediators, including JNK, MKK4, ATF2, and c-Jun [64]. This suggests that PP2A’s anti-tumor properties may involve pathways beyond ERK and Akt signaling, indicating the need for further research to fully elucidate the complex role of PP2A in tumorigenesis.

## 4. Endogenous Inhibitors of PP2A in Pulmonary Diseases

PP2A has multiple endogenous and exogenous inhibitors, but the major endogenous inhibitors are cancerous inhibitor of PP2A (CIP2A), inhibitor 2 of PP2A (I2PP2A/SET), endosulfine α (ENSA), ubiquitin E3 ligase midline 1 (MID1), and the protein phosphatase methylesterase 1 (PME-1) [35,49].

### 4.1. Cancerous Inhibitor of PP2A (CIP2A)

Our group has shown that cigarette smoke can enhance CIP2A expression resulting in PP2A inactivation [44]. CIP2A homodimer binds to PP2A heterotrimers containing B56 subunits and inhibits by ejecting the A-subunit from the heterotrimer [103]. CIP2A regulates PP2A and subsequently downstream effects on PLK1, E2F1, Akt, DAPK1, and c-MYC in multiple types of cancer [104]. CIP2A expression is upregulated by numerous factors, including microRNAs (miRNAs) and EGFR [105]. EGFR-MEK pathway utilizes the ETS1 transcription factor to induce CIP2A expression [105]. We observed elevated EGFR signaling in human bronchial epithelial (HBE) cells from COPD patients compared to cells from healthy individuals [44]. Inhibition of EGFR signaling with erlotinib suppressed CIP2A signaling and enhanced PP2A responses [44]. CIP2A expression is also known to be negatively regulated by two miRNAs, miR-375 and miR-383-5p [106,107]. Overexpression of miR-383-5p in H1299 lung adenocarcinoma cells led to G1 cell cycle arrest and apoptosis via CIP2A inhibition, while miR-375 mediated CIP2A repression and subsequent reduced production of oncoprotein MYC in oral tumors [107,108].

### 4.2. Inhibitor 2 of PP2A (I2PP2A/SET)

I2PP2A is a specific and potent inhibitor of PP2A displaying non-competitive kinetics with IC_50_ of 2nM in vitro [109]. I2PP2A (a truncated form of SET) interacts with two subunits of PP2A, the PP2A-A subunit, and the PP2A-C subunit, via direct binding to both the N and C terminal regions, thereby inhibiting PP2A activity. Modulation of I2PP2A in lung cancer cell lines modulates PP2A activity and subsequent phosphorylation of AKT and ERK, which are linked to cell proliferation [109,110,111,112]. SET also regulates cyclin D1 and p27, which are integral in the cell cycle and can regulate MMP9 secretion, contributing to the disruption of the extracellular matrix and enhancing cancer cell metastasis [112]. SET also plays a role in epithelial to mesenchymal transition (EMT) via its interaction with c-MYC [113]. SET inhibition of PP2A ultimately leads to c-MYC phosphorylation at Ser 62, which prevents degradation and oncogenic c-MYC expression [60].

### 4.3. Ubiquitin E3 Ligase Midline 1 (MID1)

MID1 interacts with the α4 regulatory subunit of PP2A and is required for its catalytic C subunit’s ubiquitin-specific modification and proteasome-mediated degradation [114]. MID1 is upregulated in mouse bronchial epithelium following inhalation of HDM, and MID1 regulates airway inflammation by limiting PP2A activity [115]. Elevated MID1 responses via TRAIL signaling inhibit PP2A activity and correlate with lung function decline in pulmonary fibrosis [51]. This MID1-mediated regulation of PP2A activity also plays a role in lung adenocarcinoma by influencing cell cycle progression, proliferation, and apoptosis [116]. Importantly, the absence of TRAIL, in combination with reduced MID1 and protected PP2A activity, does not affect type 1 IFNs during rhinovirus infection but reduces viral replication [117]. Targeting MID1 or PP2A activity directly impacted airway hyperreactivity, IL-25, IL-33, CCL20, IL-5, IL-13, NFκB activity, p38 MAPK phosphorylation, accumulation of eosinophils, T lymphocytes, and myeloid dendritic cells, and mucus-producing cells [86,115].

### 4.4. Protein Phosphatase Methylesterase 1 (PME-1)

PME-1 is upregulated in 3.1% (4/124) of lung cancer samples and is associated with PP2A demethylation at leucine 309 resulting in the inactivation of PP2A [118]. Contrastingly, recent studies suggest PME-1 methyl-esterase activity possibly protects PP2Ac from ubiquitin/proteasome degradation in embryonic fibroblasts, with PME-1 knockout mice having lower PP2A activity compared to wild-type animals [119].

### 4.5. Newly Discovered PP2A Regulators

ENSA plays a significant role in regulating mitosis by inhibiting the activity of PP2A-B55 during the M phase. Ref. [120] ENSA interacts with PP2A mainly via the A-subunit [121]. PP2A can be influenced by the immunoglobulin-binding protein 1 (IGBP1) as it interacts with the catalytic component of PP2A in small lung adenocarcinoma, and IGBP1 is universally expressed and positively correlated to poorer prognosis [122]. IGBP1 inhibition of PP2A activity also impairs erythroid differentiation by enhancing 4EBP and p70S6k phosphorylation [123].

The PP2A catalytic subunits are subject to ubiquitination and proteasomal degradation when not bound to an A or B subunit. α4 is a non-catalytically active protein that can protect PP2A activity via C subunit binding and protects the C subunits from ubiquitination. The overexpression of α4 leads to hasten resolution of stress associated with DNA damage and increased cell survival [124].

Ceramide can activate PP2A in a stereospecific manner. In acute lung injury, TNFα induction leads to enhanced NFκB and MAPK responses, and IL8 mRNA production. TNFα activates sphingomyelinase, ultimately leading to ceramide production and accumulation. This accumulation results in PP2A activation leading to decreased inflammation [125]. Additionally, ceramide can indirectly activate PP2A by binding to SET and disrupting the interaction of SET to PPP2CA [126]. The S1P analog FTY720 is structurally related to ceramide and can bind to SET to prevent inhibition of PP2A [113].

## 5. Potential Approaches to Activate PP2A

### 5.1. Indirect Activation of PP2A via Targeting Endogenous Inhibitors

Most novel approaches aim to restore PP2A activity can be classified into direct strategies to activate the PP2A holoenzyme or its subunits and indirectly by counteracting the variety of endogenous negative regulators of PP2A (see Table 2).

### 5.2. Inhibiting SET with FTY720

As already mentioned, one strategy to inhibit SET is using ceramide and its derivatives, endogenous bioactive sphingolipids that activate PP2A by disrupting SET/PP2A interaction [127,128,129]. FTY720 (Fingolimod/Gilenya^®^), an FDA-approved drug by Novartis, is a sphingosine analog with immunosuppressive actions used to treat multiple sclerosis. In vivo, FTY720 is phosphorylated to FTY720-phosphate, the active immunosuppressant compound, that later binds to the sphingosine phosphate receptor (S1P1), leading to internalization and destruction of this receptor [157]. FTY720 directly binds I2PP2A/SET at the K209/Y122 residue, inactivating SET [158]. Inactivation of PP2A via SET leads to the induction of the c-myc oncogene and subsequent suppression of the *NDRG1* tumor suppressor gene, which drives epithelial to mesenchymal transition in cancerous A549/CDDP cells and impart chemoresistance [113]. FTY720 reduced epithelial-mesenchymal transition molecular markers (Snail, N-cadherin, and vimentin), increased tumor sensitivity to cisplatin, and reduced cancer cell invasion [159]. Derivatives of FTY720, like MP07-66 and OSU-2S, demonstrated FTY720-like antiproliferative activity in human hepatocellular carcinoma while nullifying its immunosuppressive effects [131]. Whether these compounds will demonstrate improved morbidity and mortality outcomes in lung cancer is yet to be determined.

In addition to its immunosuppressive functions, FTY720 has an additional immunomodulatory role in modifying hypersensitivity response in reactive airway disease [115]. In mice challenged with HDM, allergic airway response is correlated with upregulation of MID1, inhibition of PP2A, and activation of inflammatory mediators NFκB, p38 MAPK, and JNK [115]. These changes are MIDI dependent as inhibition of MID1 using siRNA led to a rise in PP2A and dephosphorylation of NF-κB, p38 MAPK, and JNK [115]. This is important for asthma as several studies note increased p38 MAPK signaling in the epithelium of severe asthmatics [160,161], phosphorylation of NFκB leading to Th-2 cell-mediated allergic airway responses, and JNK phosphorylation regulation of glucocorticoid receptor responses in acute asthmatic exacerbation [66,162]. Treating mice with 2-amino-4-(4-heptyloyphenol)-2-methylbutanol (AAL), a non-phosphorylatable version of FTY720, before exposure to noxious stimuli reduces total lung resistance, inflammatory cells recruitment to the lung, and increased dynamic compliance [115].

### 5.3. Next-Generation SET Inhibitors

Another possible option to inhibit SET involves apolipoprotein E (ApoE), a multifunctional protein that has a role in cholesterol transport and immunoregulation [163,164,165,166]. ApoE, and apoE-mimetic peptides, COG112 and COG449 (OP449), activate PP2A by binding the C-terminal end of SET. Refs. [132,133,134] OP449 is effective when combined with tyrosine kinase inhibitors for treating acute and chronic myeloid leukemia [132]. Finally, TGI1002 is a small 2-phenyloxypyrimidine molecule that also disrupts the PP2A/SET interaction and can increase in vitro activity of PP2A [135].

### 5.4. Inhibiting CIP2A

#### 5.4.1. Erlotinib Derivatives

Two FDA-approved drugs inhibit CIP2A: erlotinib and bortezomib. Erlotinib (Tarceva^®^) is an EGFR kinase inhibitor indicated for treating EGFR-mutant NSLC [136,137]. However, several observations suggest that erlotinib may also have activity on CIP2A, independent of EGFR. Erlotinib disubstituted quinazoline derivative, TD52, inhibits CIP2A responses without the specific EGFR effects. TD52-treated triple-negative breast cancer xenografts exhibited a reduction in CIP2A signaling, tumor burden, and increased apoptosis. Notably, treatments were well tolerated without mice body weight changes. TD52 administration reduced p-Elk1 and decreased its binding to the CIP2A promoter, reducing CIP2A levels in vivo [138,139,140]. Separately, our group has demonstrated that erlotinib can suppress CIP2A expression in HBE cells isolated from COPD patients [44]. This reduction in CIP2A led to enhanced PP2A activity and reduced innate immune and protease responses.

Bortezomib (Velcade^®^), a proteasome inhibitor first approved for treating multiple myeloma, has indirect PP2A-mediated activity potential [141]. Molecular studies suggest Bortezmobib’s pro-apoptotic effects are partially mediated by induction of PP2A dephosphorylation and subsequent inactivation of known oncoprotein Akt at Ser 473 [167,168]. Treatment of hepatic adenocarcinoma and triple-negative primary ductal carcinoma cells with bortezomib in vitro showed a dose-dependent reduction in CIP2A mRNA production without affecting protein stability or proteasomal degradation [142,143]. Although the specific mechanism of CIP2A transcription suppression is yet to be reported, this is of great interest as CIP2A repression and subsequent upregulation of PP2A activity in key cancer-promoting pathways (such as Akt and c-myc) affects the sensitivity of these solid tumors to bortezomib.

#### 5.4.2. Metformin

Metformin, a frontline agent in treating insulin-resistant type 2 diabetes, can increase PP2A activity via CIP2A inhibition. While the biochemical mechanism of CIP2A suppression is yet to be determined, metformin treatment alone correlates with significantly reduced CIP2A levels via increased proteasomal degradation [144]. Furthermore, its inhibition of oxidative phosphorylation coupled with fasting-induced hypoglycemia (and subsequent suppression in glycolysis) repressed tumor growth in the human colorectal carcinoma xenograft mouse model [144]. Inhibition of GSK3β counteracts metformin’s anti-tumor effect in glucose-deprived environment [144]. GSK3β is a Ser/Thr kinase critical in regulating protein synthesis, cell growth, differentiation, and death. PP2A is a known activator of GSK3β via dephosphorylation of serine residues nine and others [169,170]. Of note, while metformin activates PP2A activity, glucose depletion promotes transcription of the *PPP2R5D* gene encoding PP2A B56δ regulatory subunit, ensuring GSK3β activating specificity. There are two clinical trials currently investigating possible therapeutic effects of fasting with metformin therapy on stage I-III triple-negative breast cancer and ductal carcinoma in situ [171,172].

Given metformin’s relatively safe adverse effect profile, many therapeutic indications for its use are being explored. Metformin’s activation of PP2A can preventing spatial memory deficits in Alzheimer’s disease rat model, possibly by avoiding tau protein hyperphosphorylation [173]. Furthermore, Katila et al. demonstrate metformin’s neuroprotective effects in mitigating 1-methyl-4-phenyl-1,2,3,6-tetrahydropyridine (MPTP) neurotoxicity via increased α-synuclein dephosphorylation in a similar manner [174]. Active clinical trials to assess Metformin use as secondary prophylaxis in amyotrophic lateral sclerosis, frontotemporal dementia, and Huntington’s disease [175,176].

#### 5.4.3. Other Newer CIP2A Inhibitors

As research in the field of CIP2A inhibition continues, new agents such as Celastrol (tripterine), a compound found in traditional Chinese medicine, and ethoxysanguinarine, a benzophenanthridine alkaloid extracted from *Macleaya cordata* that can inhibit the function of this protein, are being investigated [101,102,177,178,179].

### 5.5. Inhibiting PME-1

PME1 is a serine hydrolase that facilitates reversible demethylation and inactivation of PP2A at the leucine 309 residue [180]. Methylation of the L309 carboxyl group is critical for PP2A-B subunit binding, formation of PP2A holoenzyme assembly, and targeting specificity [181]. In conjunction with PP2A demethylation, PME-1 is believed to bind directly to the PP2A-C subunit’s catalytic domain, inducing conformational change and inactivating the enzyme. PME-1 can upregulate ERK activity and increasing its downstream target Elk-1. Enhanced Elk-1 mediated gene transcription contributes to malignant cell growth in glioblastoma cell lines [182,183].

Bachovin et al. identified several PME-1 inhibitors, including sulfonyl acrylonitrile base AMZ30 and aza-β-lactams-based ML174 [146]. The most potent PME-1 inhibitor is ML174 (also known as ABL127) and it covalently binds serine 156 of PME-1 via induction of serine nucleophilic attack on an ML174 carbonyl group within its beta-lactam ring [146]. While preclinical studies of ML174 are promising, their clinical potential has yet to be thoroughly investigated and little to no lung investigations are reported.

### 5.6. Direct Activation of PP2A

Two main classes of compounds target the scaffolding subunit A of PP2A and activate the holoenzyme: phenothiazines and small molecule activators of PP2A (SMAPs). Phenothiazines are FDA-approved medications traditionally used as potent dopamine receptor antagonists in treating various psychiatric disorders [184]. For example, chlorpromazine (thorazine) can induce PP2A activity, with concomitant dephosphorylation of PP2A-associated proteins and induction apoptosis [153]. They activate PP2A via a mechanism described by Gutierrez and colleagues, which is responsible for the anti-tumor effects shown by these drugs [147,153,185,186]. Reengineering phenothiazines led to the identification of SMAPs, which are a part of the tricyclic sulfonamide subclass of tricyclic neuroleptics [187]. Mechanistically, they stabilize PP2A ABC holoenzyme heterotrimers, which restores basal PP2A activity by directly activating PP2A [187].

SMAPs DT-061, DT-382, DT-794, and DT-1154 (also known as DBK-1154) outlined in Sangodkar et al. were tested in KRAS mutant lung cancer cells and demonstrated increased poly (ADP-ribose) polymerase cleavage, an indicator of DNA damage, and increased tumor cell death in a caspase-dependent manner [46]. In vivo experiments, treatment with SMAPs significant inhibited tumor growth and increased tumor cell death [46]. Molecularly, SMAP treatment substantially reduced phosphorylated ERK, the activation of which has been well described to promote growth and chemoresistance in various cancers [188,189,190]. Cross comparison between the SMAP treatment arm with a combination of MK2206 (AKT inhibitor) and AZD6244 (MEK inhibitor) demonstrated similar antiproliferative activity suggesting that the phosphatase effect of SMAP extends beyond ERK dephosphorylation alone. Furthermore, systemic administration of SMAP over 30 days induced no liver toxicity, mouse weight loss, behavioral changes, or increased mortality [46].

Our group has utilized DBK-1154 in a smoke-induced COPD mouse model and demonstrated reduced emphysematous airway remodeling compared to controls [43]. In the presence of cigarette smoke, administration of 1154 over two months was well tolerated, as seen by the absence of body weight and liver-to-body weight reduction [43]. Treatment with DBK-1154 reduced smoke-induced ductal destruction, immune cell infiltration, airway enlargement, and cathepsin S production [43]. A new SMAPs was recently described to be an improvement on DBK-1154, due to DBK-1154 having low stability to oxidative metabolism, resulting in fast clearance and limited systemic exposure after an oral dose [148]. This new SMAP, ATUX-792 has modifications to the carbazole tricyclic, central pyran ring constraint, and chlorine substitution of the carbazole group that makes ATUX-792 more resistance to oxidation [148,191]. Therefore, improving the bioavailability of these compounds may increase the potential for future human trials.

### 5.7. Unknown Mechanism of Targeting PP2A

A select group of compounds are reported to activate PP2A, but without a suggested mechanism. Xylulose-5-phosphate (X5P), a nucleotide precursor in the pentose phosphate pathway, added to fractionated hepatic murine lysates, increased free phosphate resulting from X5P-induced PP2A activity [149]. Carnosic acid, a polyphenolic diterpene isolated from rosemary, can inactivate intracellular signaling pathways by influencing PP2A responses in prostate cancer cells. However, it shows demethylation of PP2A with consequent inhibition of PP2A activity in skeletal muscle cells [192,193]. This discordance in the action of carnosic acid on PP2A highlights the importance of discovering the mechanisms of actions of these PP2A-modulating strategies for potential drug development approaches. Vitamin E analogs like α-Tocopheryl succinate (α-TOS) can activate PP2A [150,151]. Forskolin is a diterpenoid derived from the *Coleus forskohlii* root that also activates PP2A and treatment with it results in the dephosphorylation of PP2A substrates [152].

## 6. Potential Therapeutic Benefits of PP2A Inhibition

There are several studies suggesting that inhibition of PP2A may be beneficial. For example, knockdown of PP2A’s catalytic c-subunit α (PP2Acα) can restore glycogen production in the liver, reduce serum glucose, and reduce hepatic fibrosis resulting from repeated hepatic injury [194,195]. LB-100, derived from Cantharidin, a compound secreted from blistering beetles used in traditional Chinese medicine, can inhibit PP2Acα activity [196,197,198] by displaces one of two Mn (2+) cofactors within the PP2A C subunit [198]. Chen et al. demonstrated that six weeks of intraperitoneal injection of LB-100 significantly attenuated high-fat diet-induced hepatic steatosis and inflammation in C57BL/6 mice. Molecularly, LB-100 administration was correlated with significantly downregulated Srebp1 and downstream gene targets involved in lipogenesis [199].

Recent studies demonstrated LB-100’s therapeutic potential in various disease contexts, including sensitization of malignant meningioma to radiation therapy and enhancing T cell-mediated immunity against glioblastoma [200,201]. In 2016, LB-100 completed a phase I clinical trial for treating solid tumors with Docetaxel. The study enrolled 29 patients and demonstrated favorable safety, tolerability, and preliminary therapeutic potentials in sarcomas, thymoma, and atypical carcinoids of lung, ovarian, testicular, breast, and prostate cancer [202]. Clinical trials are ongoing to investigate the potential clinical benefits of LB-100 use in glioblastoma, metastatic small-cell lung cancer, and myelodysplastic syndrome [201].

## 7. Potential Negative Impact of Systemic Targeting PP2A

PP2A has different physiologic functions, including playing a role in the maturation of germ cells, maintaining homeostasis of various body organs, tumor suppression, and regulation of metabolic processes. Although this review primarily focuses on pulmonary functions of PP2A, we will briefly outline essential PP2A functions throughout the body as systemic treatment with PP2A activators could influence several critical responses within the body. A recent study suggests that antiphospholipid antibody recognition of β2 glycoprotein I promote thrombosis and the authors suggested that inhibition of PP2A could be a potential therapeutic mechanism for thrombosis [203].

Dysfunction of PP2A signaling can lead to cardiac hypertrophy, increased levels of atrial natriuretic peptide and B-type natriuretic peptide, ventricular fibrosis, and impairment in the cardiac contractile function in mouse studies [204,205,206]. PP2A in the myocardium dephosphorylates proteins involved in excitation-contraction coupling, which inactivates or decreases the function of the following: β-adrenergic receptor, L-type Ca^2+^ channel, ryanodine receptor, phospholamban, troponin I, myosin light chain [204]. Increased protein phosphatase activity is observed in failing human and animal hearts [207]. A study of transgenic mice transplanted with CD-1 mice overexpressing the PP2A catalytic domain was subjected to left anterior descending artery-ligation surgery or sham surgery. A month after myocardial infarction, histologic slides from the transgenic mice indicated dilated cardiomyopathy, decreased function on echocardiogram, and decreased function of sarco/endoplasmic reticulum Ca²⁺-ATPase (SERCA) and CaMKII pumps [206]. Dysfunction of PP2A units can also lead to dilated cardiomyopathy and impaired LV function [208]. Although dysregulation of PP2A pathways leads to different cardiac phenotypes, dysfunction of the Ca^2+^ response pathway might be the common pathway involved in the pathogenesis of heart failure phenotypes [209,210]. B56α subunit is a likely subunit involved in cardiac contractile function.

Recent work in patients with antiphospholipid syndrome (APS), an acquired autoimmune disease, suggest a possible association of PP2A in developing the prothrombotic state. APS pathogenesis involves an inciting factor triggering endothelial cell damage and apoptosis. This results in the formation of neoantigens formed from damaged cellular membranes bounded by serum proteins. These neoantigens sensitize an adaptive immune response by developing antiphospholipid antibodies (aPLs), mediating subsequent clinical manifestations of venous and arterial thromboembolism. Sacharidou et al.’s work on human aortic endothelial cells suggests that interaction between extracellular aPL and surface glycoprotein β2 glycoprotein I (B2GPI) culminate in downstream inhibition of eNOS, and subsequent production cessation of a key anticoagulant factor, nitric oxide, likely from PP2A mediated phosphatase activity resulting in eNOS S1177 dephosphorylation and subsequent de-activation. Of note, this investigation utilized aPL/B2GPI binding as a novel mode of PP2A activation. The group also observed that by knockdown of the PP2A-ɑ subunit (PR65ɑ), aPL suppression of eNOS was reversed. PP2A’s holoenzyme diverse heterogeneity stems from a large possible combination of variable subunits. A lack of data exists to inform whether differing modes of PP2A activation result in the common PP2A activation profile or distinct variability of PP2A activation dependent on the activation mechanism. Furthermore, whether PP2A activation contributes to systemic thrombophilia in individuals without serum antiphospholipid autoantibodies has yet to be determined [203].

Insulin sensitivity is a hallmark of various metabolic diseases, including type 2 diabetes Mellitus. PP2A mRNA levels and activity levels were elevated in the liver and muscle tissue of the insulin-resistant Zucker Diabetic Fatty (ZDF) rat model [211]. Insulin mediates activation (phosphorylation) of Akt. This leads to the inactivation of FOX01 and inhibition of gluconeogenesis [212,213]. Activation of Akt also results in the inactivation of Gsk3α and activation of glycogen synthase and resulting in increased glycogen synthesis [214]. Insulin acts through aPKCs to stimulate sterol regulatory element-binding protein 1 (Srebp1c) and increase FFA and TG synthesis [215,216]. In insulin-resistant mice, insulin activation of Akt is impaired. This is likely secondary to increased Akt-specific PP2A phosphatase activity [217,218]. Ultimately, this leads to increased serum glucose via enhanced gluconeogenesis and reduction in glycogen synthesis; given that aPKCs stimulation is not impaired, increased serum FFA and TG are also observed in the insulin-resistant state.

## 8. Conclusions

There is mounting evidence that PP2A phosphatase responses are subdued in several pulmonary diseases which could plays an important role in altered inflammation signaling. The overlapping prevalence of some of these pulmonary diseases and the possible existence of common underlying molecular mechanisms, suggest that targeting similar pathways in pulmonary diseases may be a logical approach to future treatment and indicate that PP2A may be a key pathogenic link in the progression of several pulmonary diseases. Since PP2A is a key regulator of inflammation and proteolytic responses, reduced PP2A could alter the inflammation and extracellular matrix responses in the lungs triggering altered lung function, pathology, and promote tumor initiation, growth and possibly metastasis. Currently, there are several drugs that could be utilized to exert therapeutic effects on PP2A activation, directly or indirectly. However, further extensive research is required to determine the complex biology of PP2A and possibly improve the bioavailability of current PP2A-activating compounds. Equally, systemic effects of PP2A activation needs addressing in parallel with investigating the subsequent pulmonary outcomes. Clinical trials of these compounds are likely to take place first in cancer, where the balance of risk to benefit are more pronounced, though IPF which has a prognosis, after diagnosis, similar to lung cancer may be an exception to this. Determining the upstreaming regulators of PP2A and subsequent downstream effects of PP2A signaling at the initiation and through the progression of the disease needs to be thoroughly investigated.

## Figures and Tables

**Figure 1 medicina-59-01552-f001:**
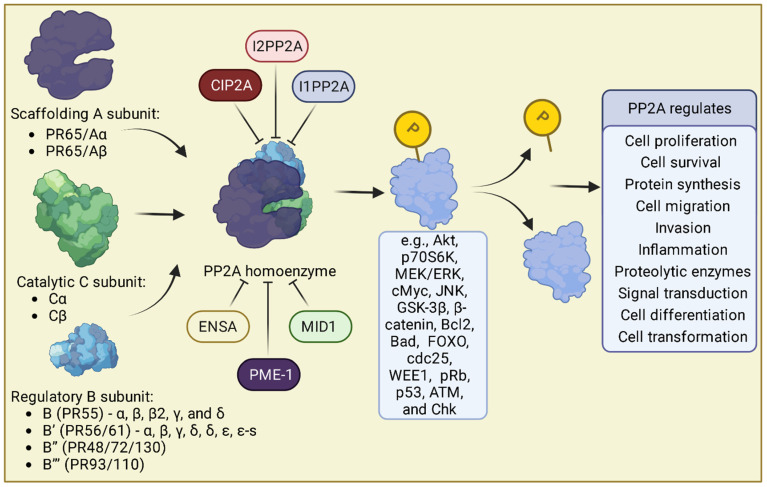
PP2A subunits, endogenous inhibitors, and common signaling pathways. Created with BioRender.com.

**Table 1 medicina-59-01552-t001:** Status of PP2A signaling in pulmonary diseases.

Disease	PP2A Status	Mechanisms for Altered PP2A	Downstream of PP2A Signaling	Reference
COPD	Reduced PP2A responses	Increased CIP2A and SET signaling, reduced antioxidant responses, viral induced suppression of PP2A, inhibition of PTP1B responses, phosphorylation of the C subunit of PP2A	Increased: Immune cell infiltration, MAPK signaling, Cathepsin S expression, phosphorylation of TTP, innate inflammation, NFκB signaling	[43,44,52,53,54,55]
AAT deficiency	Reduced PP2A responses	Loss of AAT, inhibition of PTP1B responses, reduced PKA responses	Increased: Innate immune responses, proteolytic responses	[45]
Lung cancer	Reduced PP2A responses	Loss of PP2A subunits via gene mutations, increased CIP2A expression, and EGFR mutations	Dysfunction of downstream GTPase RalA, dysfunctional cell growth, migration, and apoptosis. Increased ERK, MKK4, ATF2, and c-Jun Akt, and c-Myc signaling	[46,56,57,58,59,60,61,62,63,64]
Asthma	Reduced PP2A responses	Eosinophil peroxidase enhances PP2A phosphorylation but other mechanisms may exist for suppression of PP2A	Increased: Phosphorylation of p38, JNK-1 and GR at site Ser226, CCL4, IL-13, and iNOS expression, serum IgE levels	[65,66,67,68,69]
IPF	Reduced PP2A responses	Low α2β1 integrin receptor concentrations lead to decreased PP2A levels; elevated MID1 and TRAIL expression	Increased: Inflammation, profibrotic genes, pulmonary fibrosis, and matrix collagen deposition in mouse models	[50,51,70,71]

**Table 2 medicina-59-01552-t002:** Therapeutic targeting of PP2A.

Compound/Drug	Mode of Action	Impact on PP2A	FDA-Approval Status	Downstream Effects	Reference
FTY720	FTY720 binds I2PP2A/SET at the K209/Y122 residue, inactivating SET	Indirectly increases PP2A activity	Fingolimod/Gilenya^®^, FDA-approved drug by Novartis to treat multiple sclerosis	Suppression of c-myc and upregulation of NDRG1, Reduces EMT	[127,128,129]
FTY720 derivative: AAL(s), OSU-2S	Disruption of the SET-PP2A complex	Indirectly increases PP2A activity	Experimental use only	AAL(s) enhances TTP responses	[79,130]
FTY720 derivatives: MP07-66	Disruption of the SET-PP2A complex	Indirectly increases PP2A activity	Experimental use only	Antiproliferative activity in human hepatocellular carcinoma without immunosuppressive effects	[131]
ApoE-derived peptides: COG1410, COG112	Binding to the C-terminal end of SET	Indirectly increases PP2A activity	Experimental use only	Inhibition of Akt signaling, cellular proliferation, cellular migration, and invasion	[132,133,134]
TGI1002	Disrupts SET-PP2A interaction	Indirectly increases PP2A activity	Experimental use only	Increases dephosphorylation of BCR-ABL, inhibits tumor growth	[135]
EGFR kinase inhibitors: erlotinib	Erlotinib inhibits CIP2A responses	Indirectly increases PP2A activity	Erlotinib (Tarceva^®^) is approved for treating EGFR-mutant NSLC	Induces apoptosis in hepatocellular carcinomas, reduces smoke induced innate immune and protease responses	[44,136,137]
Erlotinib derivative: TD52	Inhibits CIP2A independently of EGFR signaling	Indirectly increases PP2A activity	Experimental use only	Reduces CIP2A signaling, tumor burden, and increased apoptosis	[138,139,140]
Proteasome inhibitor: bortezomib	Suppresses CIP2A by undefined mechanism	Indirectly increases PP2A activity	Bortezomib (Velcade^®^), approved for treating multiple myeloma	Tumor growth inhibition	[141,142,143]
Metformin	CIP2A inhibition	Indirectly increases PP2A activity	Approved as an antidiabetic agent used in type 2 diabetes mellitus	Inhibition of GSK3β, represses tumor growth, indirectly leads to dephosphorylation of many proteins	[144,145]
Celastrol (tripterine)	CIP2A inhibition through the ubiquitin-proteasome pathway	Indirectly increases PP2A activity	Experimental use only	Inhibited cell proliferation and induced apoptosis in NSCL	[102]
Ethoxysanguinarine	CIP2A inhibition	Indirectly increases PP2A activity	Experimental use only	Downregulates c-Myc and pAkt, inhibits proliferation and induces apoptosis of lung cancer cells	[101]
PME-1 inhibitors: AMZ30 and ML174	Inhibit PME-1 signaling	Reduces demethylation of PP2A, increases PP2A activity	Experimental use only	Decreases cell proliferation and invasive growth in vitro	[146]
SMAPs: DBK-1154, DT-382, DT-794, DT-061, and ATUX-792	SMAP binding stabilizes and promotes PP2A heterotrimeric holoenzyme assembly	Directly activate PP2A	Experimental use only	Increased ADP-ribose cleavage, increased tumor cell death, increase tumor necrosis, reduce cathepsin S expression, MAP kinases responses	[43,46,147,148]
Xylulose-5-phosphate	Increases free phosphate	Indirectly increases PP2A activity	Experimental use only	Possible regulation of glucose metabolism and fat synthesis	[149]
α-Tocopheryl succinate	Unknown mechanism	Unknown if direct or indirect PP2A activation	Experimental use only	Inhibition of JNK, Akt, MAPK, NFκB, Sp1 and the androgen receptor	[150,151]
Forskolin	Unknown mechanism	Unknown if direct or indirect PP2A activation	Experimental use only	Dephosphorylation of PP2A substrates such as EF-2 and RB	[152]
Chlorpromazine (Thorazine)	Same mechanism as SMAPs	Direct activation of PP2A	Phenothiazine neuroleptic, FDA approved for short-term management of severe anxiety and psychotic aggression	Dephosphorylation of multiple PP2A substrates and subsequently induces apoptosis	[153]
Salmeterol	Unknown mechanism	Unknown if direct or indirect PP2A activation	FDA approved in the management and treatment of asthma and COPD	Reduced immune cell infiltration and innate immune responses in HDM mouse model	[84]
Theophylline	Unknown mechanism	Unknown mechanism to activate PP2A but is independent of its inhibition of PDE	FDA approved for the treatment of asthma and COPD	Inhibits type III and type IV phosphodiesterase (PDE). It also binds to the adenosine A2B receptor	[82,154,155,156]

## Data Availability

Not applicable.

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
