# Peer review of "Protein Phosphatase 2A as a Therapeutic Target in Pulmonary Diseases"

_medicina, 2023, doi:10.3390/medicina59091552_

Round 1

Reviewer 1 Report

The topic of the review is relevant for the field; the manuscript is nicely wrtitten and completed by appealing figures. No further modification needed.

Author Response

We thank the reviewer for their feedback

Reviewer 2 Report

It was my pleasure to review this interesting paper " Protein Phosphatase 2A as a Therapeutic Target in Pulmonary Diseases".

Novel therapeutic strategies for treating pulmonary diseases are needed.

The authors outline the potential impact of reduced PPA2 activity in pulmonary diseases and potential therapeutic strategies.

Congratulations on well written paper.

Author Response

(The authors gave the same response as above.)

Reviewer 3 Report

Review of Manuscript ID medicina-2551178: Protein Phosphatase 2A as a Therapeutic Target in Pulmonary Diseases

It was a great pleasure for me to review this manuscript.

In this very interesting review, the authors very clearly outline the activation status of PP2A in pulmonary diseases and the resulting impact on multiple signalling pathways with particular interest in the potential development of therapeutics to restore normal PP2A responses in the lung.

Assessing the potential opportunities to target PP2A responses is a good idea and should be reviewed.

In the introduction, the authors have well presented the epidemiological problem of all lung diseases in which targeting PP2A might be of particular benefit.

In general, the review seems well structured and relevant from a scientific point of view.

However, I have one small comment that I think should be addressed more.

The author mention that theophylline enhances PP2A responses to combat inflammation in A549 cells. But in Table 2, which clearly explains the possible therapeutic effects of PP2A, theophylline is not mentioned. I suggest including theophylline as a possible therapeutic agent in Table 2, as there are other references besides those cited that need to be reviewed:

Barnes PJ. Teophylline. Pharmaceuticals 2010;3:725-747

Allen SC, Tiwari D. Teophylline as a systematic anti-inflammatory agent: the need for its revival as a possible adjunctive treatment for “Inflammaging”. Biol Eng Med 2019;4(1):1-3.

Allen S, et al. Inflammation and muscle weakness in COPD: Considering a renewed role for theophylline? Curr Resp Med Rev 2018;14(1):35-41.

Kanahera M, et al. Anti-inflammatory effects and clinical efficacy of theophylline and tulboterol in mild-to-moderate chronic obstructive pulmonary disease. Pulm Pharmacol Therap 2008;21:874-878.

Author Response

We thank the reviewer for the helpful feedback and we have added your recommended references in addition to 2 other references. Please see lines 218-219 and 259-264, and the bottom of Table 2 for these changes.